# Multi-Band High-Efficiency Multi-Functional Polarization Controller Based on Terahertz Metasurface

**DOI:** 10.3390/nano12183189

**Published:** 2022-09-14

**Authors:** Huaijun Chen, Wenxia Zhao, Xuejian Gong, Lianlian Du, Yunshan Cao, Shilong Zhai, Kun Song

**Affiliations:** 1College of Physics and Electronic Information Engineering, Engineering Research Center of Nanostructure and Functional Materials, Ningxia Normal University, Guyuan 756000, China; 2Department of Applied Physics, Northwestern Polytechnical University, Xi’an 710129, China

**Keywords:** terahertz, metasurface, polarization controller, multi-functional

## Abstract

Electromagnetic metasurfaces with excellent electromagnetic wave regulation properties are promising for designing high-performance polarization control devices, while the application prospect of electromagnetic metasurfaces is limited because of the current development situations of the complex structure, low conversion efficiency, and narrow working bandwidth. In this work, we design a type of reflective terahertz metasurface made of a simple structure that can achieve multiple polarization modulation with high efficiency. It is shown that the presented metasurface can realize ultra-broadband, cross-polarization conversion with the relative working bandwidth reaching 94% and a conversion efficiency of over 90%. In addition, the proposed metasurface can also efficiently accomplish different polarization conversion functions, such as linear-to-linear, linear-to-circular, or circular-to-linear polarization conversion in multiple frequency bands. Due to the excellent properties, the designed metasurface can be used as a high-efficiency multi-functional polarization modulation device, and it has important application value in terahertz imaging, communication, biological detection, and other fields.

## 1. Introduction

Because of its unique advantages, terahertz technology has very important applications in many fields, such as communication, detection, biological detection, material science, and so on [1,2]. The polarization state, an important attribute of terahertz wave, plays an important role in terahertz technology, and regulating the polarization state of a terahertz wave has become a popular research issue. Traditionally, people usually use metal grating structures, quartz plates, and other devices to regulate the polarization state of a terahertz wave [3,4]. However, these devices usually have problems, such as bulky, narrow bandwidth, large loss, and difficult integration. The design concept of metasurface provides a new channel for solving the above problems.

Metasurface, a kind of two-dimensional artificial material composed of subwavelength microstructures, can achieve many fantastic phenomena, which cannot be realized by natural materials, such as abnormal reflection/refraction [5,6,7,8,9], electromagnetic stealth [10,11], perfect absorption [12,13], achromatic lens [14,15], and so on. The recent progresses have demonstrated that metasurfaces can also achieve much stronger chiral or anisotropic effect than natural materials through specific structure designs [16,17,18,19,20,21], which therefore enables them to be especially suitable for designing high-performance terahertz polarization control devices [22,23,24,25]. Zhou et al. designed a terahertz chiral metasurface composed of conjugated gammadions, which realized the dynamically adjustable polarization rotation effect by combining with the photosensitive medium [26]. Cong et al. confirmed a transmission terahertz polarization controller composed of three-layer metallic grating metasurfaces, which achieved broadband 90º polarization rotation via the Fabry–Perot cavity effect, and the working bandwidth reached 0.8 THz accompanied by the conversion efficiency as high as 85% [27]. Jiang et al. proposed a method to compensate inherent resonant dispersion of metal structure by the thickness-dependent dispersion of the dielectric spacing layer and designed a birefringent metasurface, consisting of L-shaped metal structure-SiO_2_ dielectric layer-metal backboard, which could be used as a dispersion-free half-wave plate or quarter-wave plate in a broad bandwidth [28]. Quader et al. demonstrated a graphene-based metasurface that exhibited a dynamically tunable feature and could generate broadband and efficient linear-to-circular polarization conversion [29].

Despite the great progresses of the design of polarization control devices, most of the current polarization control devices usually can only achieve a single function, which extremely limits the application flexibility in fact. Recently, on the basis of function multiplexing [30], frequency multiplexing [31,32], angle multiplexing [33,34], and polarization multiplexing [30,35], multi-functional metasurfaces that can achieve multiple regulations of electromagnetic waves have been proposed. For instance, Cai et al. demonstrated a bifunctional metasurface that can simultaneously realize different functionalities of focusing and beam bending [30]. Sell et al. designed a dielectric metasurfaces that can deflect *N* different incident wavelengths to *N* unique diffraction orders [31]. More recently, the multi-functional polarization control devices, based on metasurfaces that can simultaneously realize diverse functionalities of polarization manipulation have been also proposed [36,37,38,39]. Zhen et al. proposed a fully phase-modulated terahertz metasurface that could accomplish versatile polarization conversion and wavefront shaping [40]. Based on phase change materials, Luo et al. designed a novel metasurface, with switching function, that could achieve the mutual conversion of half-wave plate and quarter-wave plate functions in the broadband range [41]. Cai et al. proposed a dynamically adjustable terahertz cascade metasurface model, which realized the simultaneous regulation of the wavefront and polarization state by rotating different metasurface layers [42]. With the development of terahertz technology, the actual demand for multi-functional terahertz polarization control devices is continuously increasing. Thus, the design of terahertz polarization devices with more functions or more working frequency bands is of great importance.

In this work, an anisotropy terahertz metasurface consisting of metal elliptical blade-polymer dielectric layer-metal back plate is proposed. The results show that the considered metasurface simultaneously possesses the functionalities of function multiplexing and frequency multiplexing. For the incident linear polarization wave, the proposed metasurface can respectively function as a half-wave plate with a polarization conversion ratio of over 90% in the ultra-broadband frequency range of 0.36–1.0 THz and at the frequencies of 1.20 THz and 1.31 THz, respectively. Meanwhile, the metasurface can also realize the linear-to-circular or circular-to-linear polarization conversion with the conversion efficiency of over 94% at frequencies of 0.32 THz, 1.03 THz, 1.18 THz, 1.21 THz, 1.29 THz, and 1.32 THz. Owing to the excellent properties, it can be used as a multi-band high-efficiency multi-functional polarization controller, which has important potential applications in many fields, such as multi-channel communication, imaging system, biological detection, and so on.

## 2. Theory and Structure Design

Figure 1 depicts the schematic geometry of the proposed terahertz metasurface. The structure parameters of the terahertz metasurface are given as: *p* = 200 μm, *r*_u_ = 240 μm, *r*_v_ = 56 μm, *t* = 65 μm, the angle between the *u*-axis and *x*-axis is 45°, the permittivity of polymer dielectric plate is 2.324 + 0.003 * *i*, the metal cladding is made of gold with a thickness of 0.2 μm and a DC conductivity of 4.56 × 10^7^ S/m. The simulations were performed using the commercial finite element simulation software CST Microwave Studio 2021 (Darmstadt, Germany). In the simulations, periodic boundary conditions are applied along the *x*-axis and *y*-axis directions. The open boundary condition is applied along the *z*-axis direction, and the linear polarization incident wave propagates along the *z*-axis direction. For the incident *x*-polarization wave (***E***_i_ = *E*_0_***e***_x_), the reflected wave is $E_{R}=E_{x}e_{x}+E_{y}e_{y}=R_{xx}E_{0}e^{j\varphi_{xx}}e_{x}+R_{yx}E_{0}e^{j\varphi_{yx}}e_{y}$, where $R_{xx}=E_{x}/E_{0}$ and $R_{yx}=E_{y}/E_{0}$ represent coefficients of the co-polarization and cross-polarization reflection components, respectively, and $\varphi_{xx}$ and $\varphi_{yx}$ represent phases of the corresponding components. When $\Delta \varphi =\varphi_{xx}-\varphi_{yx}=2n\pi -\pi /2$ or Δφ=2nπ+π/2 (*n* is an integer), if $R_{xx}=R_{yx}$, the reflected wave will be a right-handed or left-handed circularly polarized wave, respectively.

## 3. Results and Discussion

Figure 2 shows the reflection spectra and phase curves for the incident x-polarization wave. As shown in Figure 2a, the cross-polarization reflection coefficients R_yx_ are over 0.95 in the frequency range of 0.36–1.0 THz, while co-polarization reflection coefficients R_xx_ are less than 0.32, indicating that the x-polarized wave is efficiently converted to y-polarized wave in this frequency range, and the relative bandwidth reaches 94%. In addition, the same polarization conversion effect can also be observed at the frequencies of 1.20 THz and 1.31 THz. The reflection coefficients of co-polarization and cross-polarization components are the same at the frequencies of 0.32 THz, 1.03 THz, 1.18 THz, 1.21 THz, 1.29 THz, and 1.32 THz, and the phase differences between the co-polarization and cross-polarization components shown in Figure 2b are 0.5 π, −0.5 π, −1.5 π, −0.5 π, −1.5 π, −0.5 π, respectively. This means that, at the corresponding frequencies, the outgoing wave is a left-handed circularly polarized wave, right-handed circularly polarized wave, left-handed circularly polarized wave, right-handed circularly polarized wave, left-handed circularly polarized wave, and right-handed circularly polarized wave, respectively. Meanwhile, the reflection coefficients ($\sqrt{R_{xx}^{2}+R_{yx}^{2}}$) of the circularly polarized wave are over 0.97. These facts imply that the designed metasurface can realize the high-efficiency multi-band linear-to-circular polarization conversion.

In order to analysis the performances of the proposed metasurface in detail, we further use Stokes Parameters to describe the polarization states of the reflection wave [43]:(1)$$S_{0}={\left|R_{xx}\right|}^{2}+{\left|R_{yx}\right|}^{2},$$
(2)$$S_{1}={\left|R_{xx}\right|}^{2}-{\left|R_{yx}\right|}^{2},$$
(3)$$S_{2}=2\left|R_{xx}\right|\left|R_{yx}\right|\cos\left(\Delta \varphi \right),$$
(4)$$S_{3}=2\left|R_{xx}\right|\left|R_{yx}\right|\sin\left(\Delta \varphi \right),$$
where *S*_0_, *S*_1_, *S*_2_, and *S*_3_ represent the total reflection ratio of reflection wave, horizontal or perpendicular linear polarization state, +45°or −45° linear and circular polarization state, respectively. According to Stokes Parameters, the ellipticity is defined as:(5)$$=\frac{S_{3}}{S_{0}}$$
when η=1, this specific case means that the reflection wave is a left-handed circularly polarized wave. When η=−1, the reflection wave is a right-handed circularly polarized wave. The performance of the circularly polarized wave can be described by the axial ratio (AR) [43]:(6)AR=10log(tanβ)
where $\beta =\frac{1}{2}\sin^{-1}\left(\frac{S_{3}}{S_{0}}\right)$ is the ellipticity angle of reflection wave. The reflection wave can be regarded as circularly polarized wave, when the AR is below 3 dB.

Figure 3a portrays the ellipticity of the reflection wave calculated on the basis of Stokes Parameters. In Figure 3a, we can see η=1 at frequencies of 0.32 THz, 1.18 THz, and 1.29 THz. This indicates the reflection wave is a left-handed circularly polarized wave at the corresponding frequencies. However, at frequencies of 1.03 THz, 1.21 THz, and 1.32 THz, the values of η are −1, implying that the reflection wave is a right-handed circularly polarized wave at these frequencies. Figure 3b shows the calculated AR of reflection circularly polarized wave. As shown in Figure 3b, the values of AR of the reflection wave are all 0 at the frequencies of 0.32 THz, 1.03 THz, 1.18 THz, 1.21 THz, 1.29 THz, and 1.32 THz. It indicates that the reflection waves are perfectly circularly polarized waves at the corresponding frequencies. The above results further confirm that this terahertz metasurface can perfectly realize multi-band linear-to-circular polarization conversion.

In order to further study the polarization conversion performance of the designed terahertz metasurface, the polarization conversion rate is calculated. The calculation formula is as follows [44]:(7)$$PCR={\left|R_{yx}\right|}^{2}/\left({\left|R_{xx}\right|}^{2}+{\left|R_{yx}\right|}^{2}\right)$$

When PCR = 1, the linear polarization wave can be completely converted to a cross-polarization wave. However, when PCR = 0.5 and Δφ=2nπ±π/2 (n is an integer), the linear polarization wave can be completely converted to a circularly polarized wave. From Figure 4, we can see that the PCR of the cross-polarization is over 90% in the frequency range of 0.36–1.0 THz and at the frequencies of 1.20 THz and 1.31 THz, further implying that this terahertz metasurface can achieve high-efficiency cross-polarization conversion. The PCR of the sample is 0.5 at the frequencies of 0.32 THz, 1.03 THz, 1.18 THz, 1.21 THz, 1.29 THz, and 1.32 THz, and the phase differences between co-polarization and cross-polarization components at the corresponding frequencies are Δφ=2nπ±π/2 (Figure 2b). This fact confirms again that this terahertz metasurface can completely realize the linear-to-circular polarization conversion with the actual conversion efficiency (${\left|R_{xx}\right|}^{2}+{\left|R_{yx}\right|}^{2}/1$) of over 94%. According to the aforementioned results, the designed terahertz metasurface can be regarded as a high-efficiency multi-functional polarization modulation device, which can simultaneously serve as an ultra-broad band half-wave plate and multi-band quarter-wave plate. Compared with the previous polarization conversion metasurfaces [45,46,47], the well-designed metasurface have the advantages of more functionalities or operating frequency bands. Meanwhile, the geometrical structures of our metasurface are also much simpler than those of the above-referenced metasurfaces, which shows more simplicity of fabrication in high frequency range.

Due to the anisotropic property of this metasurface, it can also play the function of polarization regulation for a circular-polarization wave, and will accomplish the same regulation for the left-circular and right-circularly polarized wave [44]. For the circularly polarized wave incidence, the circular polarization reflection coefficients can be separately obtained via the linear by the following formulas:(8)$$\left(\begin{array}{cc} R_{++} & R_{+-}\\ R_{-+} & R_{--} \end{array}\right)=\frac{1}{2}\left(\begin{array}{cc} R_{xx}+R_{yy}+i(R_{xy}-R_{yx}) & R_{xx}-R_{yy}-i(R_{xy}+R_{yx})\\ R_{xx}-R_{yy}+i(R_{xy}+R_{yx}) & R_{xx}+R_{yy}-i(R_{xy}-R_{yx}) \end{array}\right)$$
where the subscripts ‘+’ and ‘−’ represent the clockwise and counterclockwise circularly polarized waves when observed along +z direction, respectively. Due to the structure design of our metasurface, the co-polarization and cross-polarization reflection coefficients for the normally incident x-polarized and y-polarized waves will satisfy the following relationships: $R_{xx}=R_{yy}$, $R_{yx}=R_{xy}$ [44,45]. Consequently, Equation (8) can be further simplified as:(9)$$\left(\begin{array}{cc} R_{++} & R_{+-}\\ R_{-+} & R_{--} \end{array}\right)=\left(\begin{array}{cc} R_{xx} & -iR_{yx}\\ iR_{yx} & R_{xx} \end{array}\right)$$

It should be noted that the chirality of the circularly polarized wave will be changed when it is reflected from the interface. Thus, the reflection coefficients are defined as $R_{LR}=R_{++}$, $R_{RR}=R_{-+}$, $R_{LL}=R_{+-}$, $R_{RL}=R_{--}$. Finally, the reflection coefficients of the circularly polarized wave satisfy $R_{LL}=R_{RR}=R_{xy},\;R_{RL}=R_{LR}=R_{xx}$. That is to say, the co-polarization (cross-polarization) reflection spectra of the circularly polarized wave will be the same as the cross-polarization (co-polarization) reflection spectra of the linearly polarized wave.

Figure 5 shows the result for incident right-handed circularly polarized wave. As shown in Figure 5a, when the right-handed circularly polarized wave is incident on the metasurface, the coefficients of the co-polarization reflection component *R_RR_* are over 0.95 in the frequency range of 0.36–1.0 THz and near the frequencies of 1.20 THz and 1.31 THz. While the coefficients of the cross-polarization reflection component R_LR_ are less than 0.32 with the polarization conversion efficiency being less than 10% (Figure 5b). These facts indicate that the reflection wave is still right-handed circularly polarized wave at the corresponding frequency range. Thus, the designed metasurface can be regarded as a chiral preserving meta-mirror. The reason for generating the above phenomenon is that the rotation direction of the circularly polarized wave will change when it is reflected at the interface. That is, right-handed (left-handed) circularly polarized wave will be converted to left-handed (right-handed) circularly polarized wave. Since the metasurface can function as a half-wave plate at the corresponding frequency, it can also convert the right-handed (left-handed) circularly polarized wave to a left-handed (right-handed) circularly polarized wave. The above two factors result in the chirality of incident circularly polarized wave unchanged after being reflected from metasurface. Additionally, as the metasurface can also act as a quarter-wave plate at frequencies of 0.32 THz, 1.03 THz, 1.18THz, 1.21 THz, 1.29 THz, and 1.32 THz, the incident circularly polarized wave will thus be converted to linear polarization wave.

In Figure 6, we illustrate that the reflection spectra of the designed metasurface varies with the incident angle α for the linearly polarized incident wave. It is seen that, for the cross-polarization conversion, both the operating bandwidth and reflection coefficients decrease gradually as α increases. In spite of the decrement, the relative operating bandwidth and reflection coefficients of cross-polarization conversion remain over 65.4% and 0.85, respectively, even α rises up to 40°. For the case of linear-to-circular polarization conversion, it is obvious that the working frequencies in a high frequency region show a significant red-shift with the increment of α, while a very slight blue-shift occurs in the working frequency in the low frequency range (approximately 0.32 THz). Despite the shifts of working frequencies, multi-band linear-to-circular polarization conversion function can still be maintained regardless of the incident angle. These facts reveal that the proposed metasurface can also operate at an oblique incidence, showing more application flexibility.

To elucidate the physics mechanism of the high-efficiency polarization conversion effect of the proposed terahertz metasurface, further simulations are carried out. As shown in Figure 1, according to the design of the unit cell, it is obvious that the *u*-axis and *v*-axis are the two principal axes of the metasurface. Figure 7 shows the simulated results of incident linearly polarized wave with the polarization direction along the *u*-axis and *v*-axis, respectively. As shown in Figure 7a, the co-polarization reflection spectra, *R_vv_* and *R_uu_*, are approximately equal in the whole frequency range except for the frequency of 1.21 THz, and the reflection coefficients are very close to 1. It can be seen in Figure 7b that the phase difference Δφ between the two major axes approximately maintains ± π in the frequency range of 0.36–1.0 THz and at the frequencies of 1.20 THz and 1.31 THz. Due to this feature, the designed metasurface can function as a high-efficiency half-wave plate. In addition, as the phase difference between two major axes is Δφ=2nπ±π⁄2 (n is an integer) at the frequencies of 0.32 THz, 1.03 THz, 1.18 THz, 1.21 THz, 29 THz, and 1.32 THz, the metasurface can therefore also achieve the function of multi-band quarter-wave plate.

Actually, the polarization conversion mechanism of the designed metasurface can be explained deeply via the interference effect in the Fabry–Pérot-like cavity formed by metal elliptical blade array and metal ground plane. As an *x*-polarized plane wave with electric vector ***E***_0_ impacts on the metasurface, it will excite dipole oscillation ***p*** primarily along the major axis (*u*-axis) of metal elliptical blade, which can be decomposed into two oscillations *p_x_* and *p_y_* along the two orthogonal directions. In addition, the co-polarized scattered field is determined by ***E***_0_ and *p_x_*, while the cross-polarized scattered field depends on *p_y_*. The multiple reflection caused by the Fabry–Pérot-like cavity may result in the interference effect of polarization couplings, which may enhance or weaken the whole co-polarized and cross-polarized reflected field. The Fabry–Pérot-like cavity with suitable length may induce a constructive interference for the cross-polarized field, with a destructive interference for the co-polarized field, and the cross-polarization conversion effect can thus be obtained. Similarly, the linear-to-circular polarization conversion can also be achieved via a Fabry–Pérot-like cavity with a certain length.

The multiple reflection process of electromagnetic wave propagating in the Fabry–Pérot-like cavity is shown schematically in Figure 8a. At the interface between air and dielectric spacer (i.e., the metal elliptical blade array), the incident wave is partially reflected to air or propagates to metal ground plane with complex propagation phase *n*_0_*k*_0_*d*, where *n*_0_ is the refractive index of dielectric spacer, *k*_0_ is the wave number in free space, and *d* is the thickness of dielectric spacer. The total reflected field can be expanded by $E_{r,\sigma }=\sum_{j=1}^{\infty }E_{r,\sigma j}(\sigma =x,y)$, where *j*−1 denotes the roundtrip within the dielectric spacer. The *m*-th term is given by [48]:(10)$$E_{r,\sigma j=m}={\left(-1\right)}^{m-1}E_{i,x}\sum_{\sigma^{\left(1\right)}\sigma^{\left(2\right)}\cdots \sigma^{\left(m-1\right)}}t_{1\sigma ;2\sigma^{\left(m-1\right)}}r_{2\sigma^{\left(m-1\right)};2\sigma^{\left(m-2\right)}}\cdots r_{2\sigma^{\left(2\right)};2\sigma^{\left(1\right)}}t_{2\sigma^{\left(1\right)};1x}e^{i2mn_{0}k_{0}d}$$
where *t_Bσ;Aσ’_* and *r_Bσ;Aσ’_* represent the transmission and reflection coefficients when wave is incident from medium ***A*** with polarization *σ’* and propagates to medium ***B*** with polarization *σ*. Figure 8b depicts the simulated and calculated results of the proposed metasurface. The theoretical calculation results on the basis of tracking the various Fabry–Pérot-like scattering processes show excellent agreement with the simulated in addition to the slight deviations, demonstrating the interference effect presented above. Therefore, we can conclude that the polarization conversion functions of the designed metasurface indeed originate from the interference effect of Fabry–Pérot-like cavity.

## 4. Conclusions

In this work, we propose a terahertz metasurface that possesses strong anisotropy. The electromagnetic characteristics of the metasurface are studied by numerical simulation, and its physical mechanism is analyzed theoretically. The results indicate that the designed metasurface can efficiently manipulate the amplitude and phase of the terahertz wave simultaneously. Additionally, the metasurface can achieve multiple polarization controlling functions, such as multi-band cross-polarization conversion, chiral preserving meta-mirror, linear-to-circular or circular-to-linear polarization conversion, and so on. Furthermore, the polarization conversion efficiency is of over 90%. Because of the excellent multi-functionality mentioned above, the proposed metasurface can be regarded as a high-performance multi-functional wave plate, which is promising for potential applications in terahertz technology.

## Figures and Tables

**Figure 1 nanomaterials-12-03189-f001:**
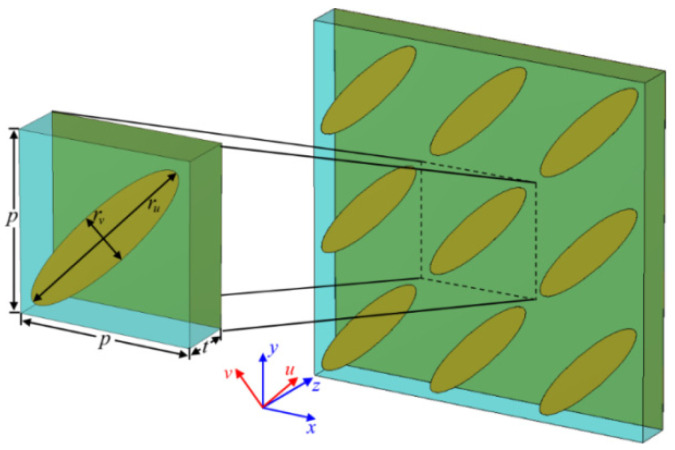
Schematic diagram of the terahertz metasurface.

**Figure 2 nanomaterials-12-03189-f002:**
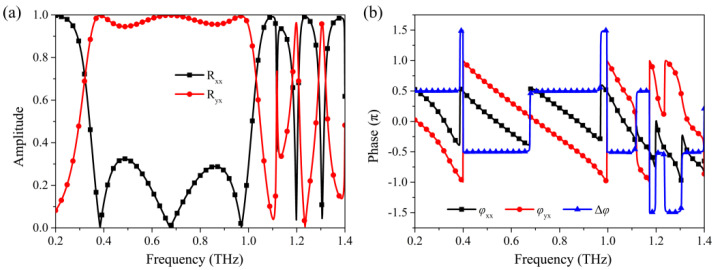
(**a**) The reflection spectra and (**b**) phase curves of the designed metasurface for incident *x*-polarized wave.

**Figure 3 nanomaterials-12-03189-f003:**
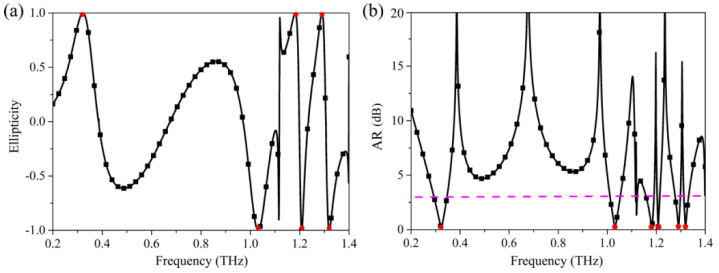
(**a**) The ellipticity and (**b**) AR of the designed metasurface for incident x-polarized wave. The purple dashed line denotes the AR of 3 dB.

**Figure 4 nanomaterials-12-03189-f004:**
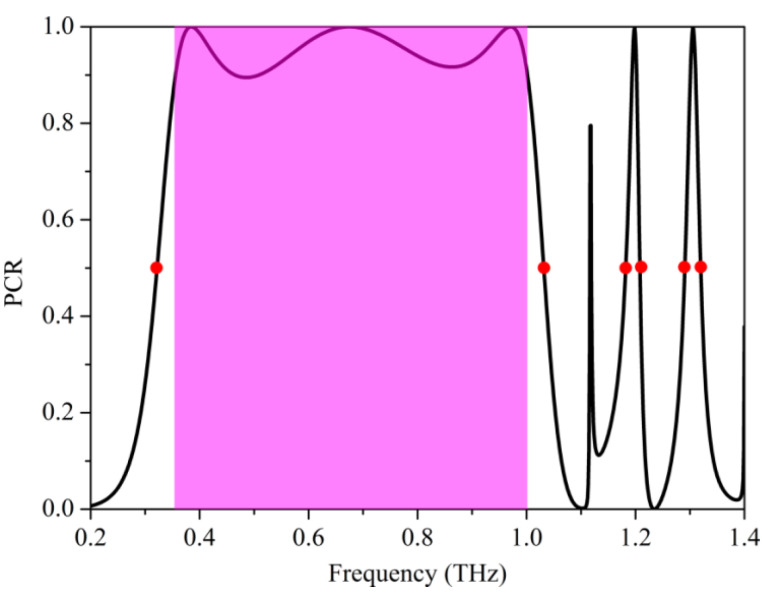
The polarization conversion efficiency of terahertz metasurface. The purple region indicates the PCR greater than 90%. The red dots denote the frequencies of 0.32 THz, 1.03 THz, 1.18 THz, 1.21 THz, 1.29 THz, and 1.32 THz, respectively.

**Figure 5 nanomaterials-12-03189-f005:**
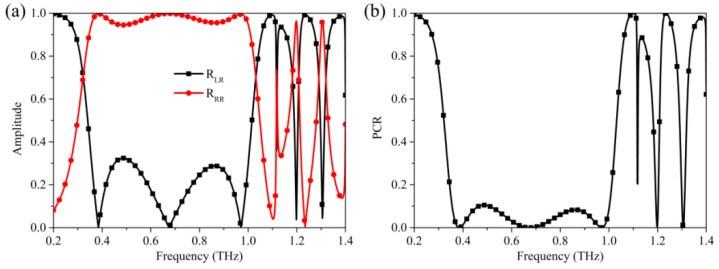
(**a**) The reflection spectra and (**b**) polarization conversion efficiency for the incident right-circularly polarized wave.

**Figure 6 nanomaterials-12-03189-f006:**
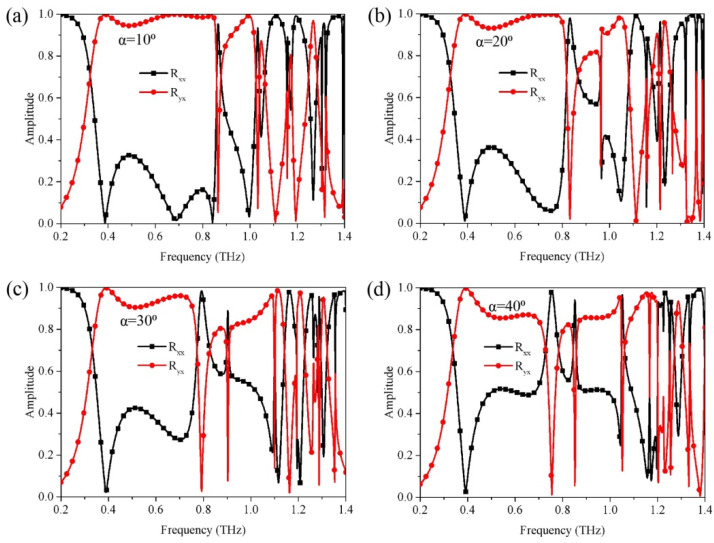
The reflection spectra of the designed metasurface vary with the incident angle for the linear polarization incidence. (**a**) *α* = 10°, (**b**) *α* = 20°, (**c**) *α* = 30°, (**d**) *α* = 40°.

**Figure 7 nanomaterials-12-03189-f007:**
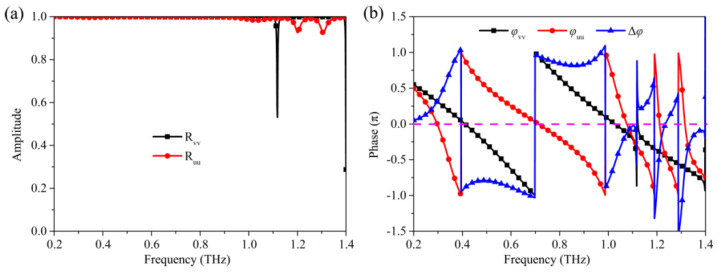
The simulation results of incident *u*- and *v*-polarization wave. (**a**) Reflection spectra, (**b**) Phases.

**Figure 8 nanomaterials-12-03189-f008:**
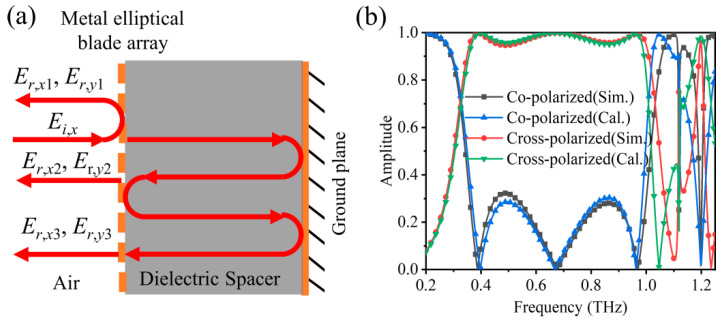
(**a**) Schematic of multiple reflection in the Fabry–Pérot-like cavity. (**b**) Simulation and calculation results of reflection spectra of the designed metasurface.

## Data Availability

All data presented in this study are available upon request from the corresponding author.

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
