# Peer review of "Multi-Band High-Efficiency Multi-Functional Polarization Controller Based on Terahertz Metasurface"

_nanomaterials, 2022, doi:10.3390/nano12183189_

Round 1
Reviewer 1 Report
The paper is dedicated to multiband linear-to-circular polarization conversion in reflection mode at terahertz frequencies. The paper can be interesting for reader but needs a significant revision.
1) The authors highlight multifunctionality of their device and cite some papers on multifunctional metasurfaces. The obtained results should be discussed, however, from a wider multifunctionality perspective. The recently proposed structures may offer multifunctional operation scenarios that include the ones with different functionalities or different manifestations of the same functionality in the neighboring frequency ranges (e.g. Adv. Opt. Mater. 5, 1700645, 2017), at different incidence angles (e.g., Sci. Rep. 7, 12228, 2017), at both frequency and angle (e.g., Nanophoton. 9, 4589, 2020; J. Appl. Phys. 131, 220901, 2022), and finally at different polarization states (e.g., Adv. Opt. Mater. 5, 1600506, 2017; Phys. Rev. Lett. 118, 113901, 2017). It is recommended to revisit the obtained results from this perspective. It is recommended to briefly discuss and cite the above-mentioned papers in order to properly indicate the role of the presented results for the multifunctionality area.
2) In line with the said above, the authors should better explain to wide reader why they consider the suggested device as a multifunctional one.
3) There may be some doubts in that the novelty is sufficient. Generally, this is not a big deal to obtain such regimes like in Figs. 2 and 5, in refelection mode. For instance, see J. Appl. Phys. 121, 174902, 2017; J. Appl. Phys. 115, 154504, 2014; Sci. Rep. 9, 4552, 2019. Therefore, the authors should better justify the proposed design and clearly show the advantages over the earlier suggested designs.
3) Notably, the microsized, not a nanosized structure is proposed, so the choice of Nanomaterials as the target journal should be justified.
4) As far as linear-to-circular polarization conversion (and probably circular-to-circular polarization conversion) looks like the main result of the paper, it is unclear why we may need PCR for linear-to-linear polarization that is given by Eq. 7?
5) It should be explained why Fig. 2 (linear-to-linear) and Fig. 5 (circular-to-circular) look similarly. What is the physics behind? Again, it looks like the novelty can be insufficient.
6) The role of the results in Fig. 7 should be better explained. It is unclear how they help to achieve the purposes of this study.
Reviewer 2 Report
The paper proposes a terahertz metasurface consisting of a metal elliptical blade - a polymer dielectric layer - a metal back plate.
Metal elliptical blades ensure the appearance of a cross-polarized wave. The metal back plate has a reflection coefficient close to 1 in the entire 0.2 – 1.4 GHz frequency range. This makes it possible to control polarization.
The results are given for normal wave incidence. However, in a metasurface working on reflection, it is desirable to separate the incident wave and the reflected wave. Therefore, it is desirable to give at least one characteristic when the wave is falling obliquely.
